# Exploring the Driving Factors Influencing Designers to Implement Green Design Practices Based on the DANP Model

**Gang Wang [1], Qigan Shao [2], Changchang Jiang [2] and James J. H. Liou [3,*]**

[1]  School of Design Art, Xiamen University of Technology, Xiamen 361024, China; 2010110505@xmut.edu.cn
[2]  School of Economics & Management, Xiamen University of Technology, Xiamen 361024, China; qgshao@xmut.edu.cn (Q.S.); ccjiang630@163.com (C.J.)
[3]  Department of Industrial Engineering and Management, National Taipei University of Technology, Taipei 10608, Taiwan
[*]  Correspondence: jamesjhliou@gmail.com

**Abstract:** Green design is a key step in improving the green performance of corporate projects. Stimulating the green design behavior of designers is the guarantee for the sustainable implementation of green design. This study extracted four dimensions, namely, external motivation, corporate-level drivers, product-level drivers and designers' attributes, and 18 indicators to consider designers' green design driving force through the literature. The DANP model was used to analyze the relationship between the indicators and the degree of importance of the indicators. The results indicated that external motivation and designer attributes influenced corporate-level drivers, while product-level drivers were outcome factors. Corporate reputation, organizational strategy and institutional pressure were the three most important criteria. Enterprises' incentives and personnel care for designers are crucial for promoting designers' continuous participation in green design.

**Keywords:** driving factors; green design; designer; DANP; green practice

## 1. Introduction

Environmental deterioration and energy shortages have threatened the sustainable development of society [1]. These problems have prompted the development of green and low-carbon products to facilitate product reuse and resource recovery. Consumers' green shopping activities have also encouraged the market to continue to develop green products, such as new energy vehicles and degradable plastic bags. Green design is a key step in the realization of green products. Thus, integrating green principles into product design is a core factor in moving toward an increase in green practices [2]. Green design systematically considers the impact on resources and the environment caused by the selection of raw materials, production, sales, use, recycling and disposal at the product design and development stages [3]. Here the characteristics of green design include eco-design, design for the environment, environmentally conscious design and sustainable design, which requires not only the harmonious development of people and the environment but also the sustainability of society, culture, consumption patterns and lifestyles. The goals of green design are to minimize the consumption of resources, to use as few as possible or no raw materials containing toxic and hazardous substances, and to reduce the generation and discharge of pollutants, all of which are practical steps toward environmental protection [4]. Designers who put green development concepts into practice and contribute to this goal are key players in and occupy a highly decisive position with respect to advancing the green development agenda [5]. That is, the participation of designers in green product design plays a vital role in the improvement of companies' green performance.

Designers' participation in the design of green products is affected by factors on multiple levels. Enterprise-level drivers, such as economic incentives and organizational

support, are important factors that motivate designers to implement green practices [6]. Murtagh et al. noted that governmental policy, public recognition and clients' requirements are motivating factors for designers to pursue green design [7]. Zhang et al. argued that market demand, corporate reputation and designer technology are the main reasons for companies to encourage designers to develop green projects [8]. In addition, the personal attributes of the designer often also affect the implementation of green practices. The designers' ability, educational background, interpersonal relationships, family members and other factors also affect their intention to implement green design [9]. Darko et al. developed a five-level framework that includes external drivers, corporate-level drivers, property-level drivers, project-level drivers and individual-level drivers to explore the factors motivating designers to participate in green projects [5]. However, few studies have considered the relationships between these motivations and the priority of these driving forces. Therefore, the purpose of this study was to determine the indicators that influence designers to implement green design practices and identify the causal relationships between indicators, as well as the degrees of importance of the indicators.

The task of investigating the driving factors that influence designers to implement green design practices relates to multiple dimensions and is obviously a multiple-criterion decision-making (MCDM) issue. The main components considered when employing the MCDM model to find the main factors that influenced designers to implement green practices were as follows:

(1)  Developing an indicator system to measure the forces motivating designers to implement green practices.
(2)  Clarifying the causal influence relationships between dimensions.
(3)  Obtaining the weights of the dimensions and indicators.

As a branch of the research field of operations research, the multiple criteria decision-making (MCDM) method is committed to quantifying the weights of evaluation indicators, ranking the importance of indicators and evaluating the performances of alternative solutions [10]. MCDM contains a cluster of technical models and scholars are constantly proposing new solutions. The Decision Making Trial and Evaluation Laboratory (DEMATEL)-based analytical network process (DANP) method is one example of an MCDM model. This method was employed to determine the impact relationships between indicators and to identify important indicators, and it was applied in many contexts, such as e-store businesses [11], stock selection [12], emerging technology promotion [13] and farm site selection [14]. To the best of our knowledge, the literature concerning the use of an MCDM model to explore the factors that influence designers to implement green projects is still rare. To fill this gap, this study applied the DANP model to the tasks of clarifying the relationship between the driving factors that influence designers to participate in green design and determining the important influencing factors.

The rest of this work is organized as follows: the literature concerning the forces motivating designers to implement green practices is discussed in Section 2. The research method is described in Section 3. The results of the analysis are presented in Section 4. The discussion is presented in Section 5. Our conclusions and possible future research directions are presented in Section 6.

## 2. Literature Review

An overview of enterprises' green practices is presented in Section 2.1. Then, the forces motivating designers to participate in green design are highlighted. Finally, an evaluation system is introduced in Section 2.3.

### 2.1. Enterprises' Green Practice

Greenhouse gases have caused climate warming and threatened global ecological security. Carbon neutrality has become a global value. Green practice is an important strategy to achieve high-quality and sustainable development for enterprises and has important implications concerning whether the goal of carbon neutrality can be achieved. The

COVID-19 outbreak has caused people to pay more attention to green consumption, and green products are gaining market power [15]. Green practice strategies worldwide are considered to be very important issues in the context of each business practice as a result of their positive impact on the economy, society, the environment and business [16]. Green design, green manufacturing, green logistics, green marketing, green 4R (reduce, reuse, recycle and recover) practices and so on are important green practices [17]. Hassan and Jaaron argued that green manufacturing practices can improve the level of total quality management and organizational performance [18]. Agyabeng-Mensah et al. proposed that green logistics management practices can promote environmental sustainability and improve the efficiency of cleaner production [19]. Tsai et al. believed that green marketing could lead to purchasing and repurchasing decisions, and they proposed a hybrid model to obtain the weights of each evaluation indicator [20]. The 4R (reduce, reuse, recycle, recover) framework is considered to be an important principle for companies to implement green practices and an important indicator for companies to achieve long-term environmental sustainability [21]. Green design has been widely identified as the greatest contributor to green development in the engineering industry [2]. Green design can change the environmental characteristics of a product or project to the greatest extent possible and, to a certain extent, determines the life cycle of the product [22]. Green design is a process that comprehensively considers environmental factors and is considered to be one of the most important aspects of sustainable development in the design industry.

*2.2. The Driving Force of Green Design*

From the perspective of technical application, green design involves renewable design technology, detachable technology design and design technology for life cycle assessment. New products can be generated by applying renewable design techniques to parts and materials of used or discarded products. The disassembly of the product is the premise of product recycling, which directly affects the recyclability of the product. How to conveniently and effectively evaluate the impact of products on people and the environment is the key to the adoption of green product design schemes by designers. Designers are participants in and executors of green design practice. They actively put green concepts into practice, thus facilitating higher energy efficiency and reducing carbon emissions. "Designer" is a general term for people who design things. Designers are usually persons who create or produce creative work in a specific specialized field and who are engaged in a combination of art and commerce. They usually use paintings or various other means of visual communication to express their work. Designers, such as industrial designers, architects, brand designers, image designers and packaging designers, are heavily involved during the initial product concept abstraction and product design development phases of green projects. Designers' green design behavior greatly affects companies' green practice performance [23]. Thus, this study focuses on research concerning the factors that encourage designers to engage in green design, with a view to increasing the enthusiasm of designers toward participating in green design to improve the performance of enterprises regarding implementing green strategies.

Green design practice includes many complex requirements and rules [24]. Designers need to integrate a great deal of professional knowledge and technological capability to meet the requirements of stakeholders (government, consumers, partners, etc.). The organizational system and corporate culture involved also have a large impact on the development of designers' creativity [25]. Cooperation between designers and negotiation between designers and partners are also important factors in the sustainable operation of green projects. Meanwhile, personal factors, such as a designer's knowledge, personality and design ability, also affect their green practice [2]. However, the existing research is not sufficient to fully understand a designer's green motivation. Analysis of the mutual influence and interactions between multiple motivations is still very rare regarding understanding green design practices. Therefore, to fill this research gap, it is necessary to

analyze the multilevel motivations of designers to implement green practices from the perspective of decision management.

### 2.3. Evaluation Framework

Determining the relevant driving factors that motivate designers to engage in green design is the first step in developing and evaluating designers' green practice models. This study used literature reviews and experts' discussions to determine the factors that affect designers' green practices. Given the initial factors extracted from the literature review, a survey was conducted with experts to determine whether these factors are considered essential factors for driving green design. Furthermore, they were asked to suggest additional deriving factors that are perceived to be relevant according to their experience. After three rounds of discussions, four dimensions and 18 criteria were selected for the final evaluation. The specific dimensions and indicators are described in Table 1.

**Table 1.** Indicators and references for the green design evaluation framework.

| Dimensions | Criteria | Explanation |
|---|---|---|
| External motivation ($D_1$) | Government policy ($C_{11}$) | The government formulates relevant green design policies to encourage enterprises to implement green design strategies. |
| | Market demand ($C_{12}$) | Adopting sustainable green design can improve product competitiveness and market share. |
| | Public recognition ($C_{13}$) | Consumers' conception of green consumption has been significantly improved, and they tend to buy products with green design attributes. |
| | Design trends ($C_{14}$) | Designers have increased their green design awareness and green design capabilities, and green design has become a design idea and design method to which the design industry has paid a great deal of attention. |
| | Industry organizational support ($C_{15}$) | The promotion and support of green design by associations, societies and other industry organizations and the recognition of green design via design awards. |
| Corporate-level drivers ($D_2$) | Organizational strategy ($C_{21}$) | The enterprise establishes a certain organizational structure and uses a systematic method to promote the implementation of green design within the enterprise. |
| | Institutional pressure ($C_{22}$) | The incorporation of the implementation of green design into project evaluation indicators. |
| | Corporate reputation ($C_{23}$) | Green design can enhance a company's industry and social influence. |

**Table 1.** *Cont.*

| Dimensions | Criteria | Explanation |
|---|---|---|
| Product-level drivers ($D_3$) | Manufacturing process and surface treatment process ($C_{31}$) | The manufacturing process and surface treatment process meets relevant environmental protection requirements and qualify as green manufacturing processes. |
| | Product standards ($C_{32}$) | Products comply with market environmental standards and green design standards. |
| | Green materials and technology ($C_{33}$) | The use of environmentally friendly materials and technologies comply with relevant environmental requirements and green manufacturing processes. |
| | Package design ($C_{34}$) | The packaging uses environmentally friendly materials and technologies to reduce transportation and storage costs and to facilitate recycling. |
| | Overall design plan ($C_{35}$) | The overall design is scientific, standardized and green. |
| Designer's attributes ($D_4$) | Designers' abilities ($C_{41}$) | The accumulation of green design knowledge, methods and experience to enhance green design capabilities. |
| | Personal norms and attitudes ($C_{42}$) | Recognition of green design ideas and concepts and mastery of green design standards. |
| | Interpersonal relationships ($C_{43}$) | Designers extensively exchange green design concepts, methods, experiences, etc. |
| | Legal awareness ($C_{44}$) | Compliance with corresponding green specifications and environmental protection standards of national and international regulations. |
| | Educational background ($C_{45}$) | The designer's level of education and knowledge of theories and methods related to green design. |

## 3. Methodology

As an MCDM method for determining the causality and influence weights within a system, DANP is the perfect combination of the Decision-Making Trial and Evaluation Laboratory (DEMATEL) and the analytic network process (ANP). Integrating the advantages of these two methods, the optimal solution to the multifactor complex relationship was obtained. First, the DEMATEL method was used to assess the relationships between the criteria. The original data were collected using a questionnaire survey, and the impact degrees, affected degrees, centralities and cause degrees of the influencing factors were calculated. Then, the ANP method was used to determine the weights of the criteria [26]. Figure 1 depicts the process of data processing using the DANP method.

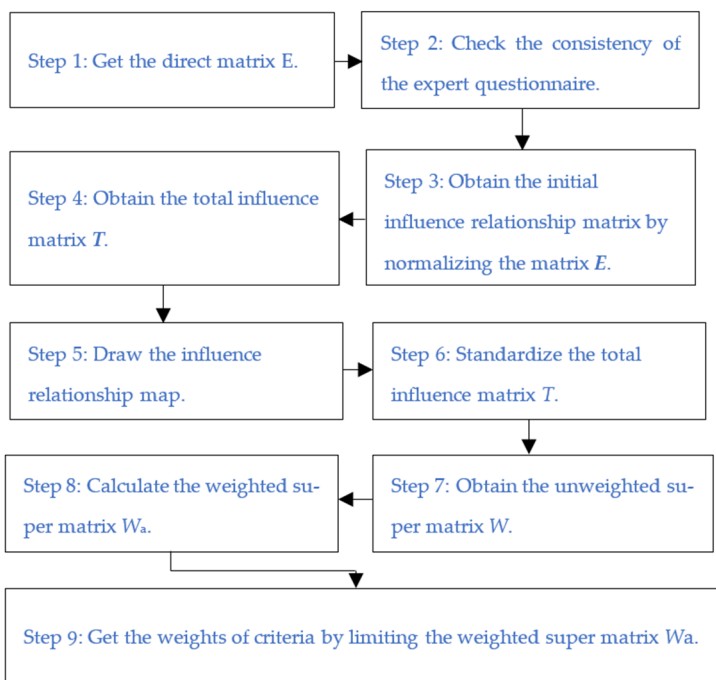

**Figure 1.** The process of data processing using the DANP method.

The specific steps are expressed as follows, and a detailed introduction to the DANP method can be found in the literature [27–30].

Step 1: The direct influence matrix $E$ was obtained.

According to the constructed criteria model, five levels of the questionnaire were included, ranging from "no influence (0)" to "extreme influence (4)". Subsequently, $k$ experts' questionnaire data were processed. As seen in Equation (1), $e_{ij}$ represents the influence degree of criterion $i$ on criterion $j$, and the average matrix $E$ with dimensions $n \times n$ was constructed as follows:

$$E = \begin{bmatrix} e_{11} & e_{12} & \cdots & e_{1n} \\ e_{21} & e_{22} & \cdots & e_{2n} \\ \vdots & \vdots & \cdots & \vdots \\ e_{n1} & e_{n2} & \cdots & e_{nn} \end{bmatrix} \tag{1}$$

Step 2: Examine the consistency of the original data.

The consensus of the $k$ questionnaires was calculated using Equation (2). If the value of the *ratio* was less than 0.05, the expert's opinion was considered to be consistent [31].

$$ratio = \frac{1}{n(n-1)} \sum_{i=1}^{n} \sum_{j=1}^{n} \left( \left| e_{ij}^{k} - e_{ij}^{k-1} \right| / e_{ij}^{k} \right) \tag{2}$$

Step 3: Obtain the initial influence relationship matrix $Y$ by normalizing the direct influence matrix.

Each row sum and column sum were calculated, the maximum was selected, each influencing factor criterion in the direct influence matrix was divided by the maximum value and the initial influence relationship matrix $Y$ was obtained, as in Equations (3) and (4).

$$Y = v \cdot E \tag{3}$$

$$v = \min \left[ \frac{1}{\max_i \sum_{j=1}^{n} |e_{ij}|}, \frac{1}{\max_j \sum_{i=1}^{n} |e_{ij}|} \right] \tag{4}$$

Step 4: The total influence matrix *T* was obtained.

Next, the indirect relationship between each criterion was analyzed, and the indirect relationships of each criterion were expressed through direct influence matrix *Y* multiplication. Then, by summing all the indirect influences, the total influence matrix *T* was calculated as follows:

$$T = Y + Y^2 + \cdots + Y^K = Y(I - Y)^{-1} \tag{5}$$

where $T = \left[t_{ij}\right]_{n \times n}, i, j = 1, 2, \ldots, n$.

Step 5: The influence relationship map (INRM) was drawn according to the total influence matrix *T*.

By analyzing each element $t_{ij}$ in matrix *T*, the mutual influence relationship between each element was calculated and described using the impact degree, affected degree, centrality and cause degree. The impact degree is the combined influence of each row of matrix *T* on all other elements, which is each row's sum ($\sum_{j=1}^{n} t_{ij}$). The affected degree refers to the combined influence value of all other elements for each column of matrix *T*, which is each column's sum ($\sum_{i=1}^{n} t_{ij}$). Centrality is the sum of the impact degree and affected degree. The cause degree is the difference between the impact degree and the affected degree. With centrality as the abscissa and cause degree as the ordinate, the INRM of the criteria model was derived.

Step 6: The total influence matrix *T* was standardized.

The effects of the primary and secondary criteria matrices must be standardized. For the primary dimension criteria influence matrix $T_D$, each element was divided by the sum of all elements in its corresponding row, as shown in Equation (7). The corresponding standardized primary dimension criteria influence matrix $T_D^a$ was obtained. To acquire the secondary dimension criteria standardized total influence matrix $T_d^a$, given that the primary dimension criteria weight was different, first, the total influence matrix *T* was divided into different dimensions of a submatrix and then calculated in terms of the unit of a submatrix, as in Equations (6) and (7).

$$T_D^a = \left[t_D^{aij}\right]_{n \times n} = \begin{bmatrix} t_D^{11}/d_1 & \cdots & t_D^{1j}/d_1 & \cdots & t_D^{1n}/d_1 \\ \vdots & & \vdots & & \vdots \\ t_D^{i1}/d_i & \cdots & t_D^{ij}/d_i & \cdots & t_D^{in}/d_i \\ \vdots & & \vdots & & \vdots \\ t_D^{n1}/d_n & \cdots & t_D^{nj}/d_n & \cdots & t_D^{nn}/d_n \end{bmatrix}, d_i = \sum_{j=1}^{n} t_D^{ij}, i = 1, 2, 3, \ldots, n \tag{6}$$

$$T_d^a = \begin{bmatrix} T_d^{a11} & \cdots & T_d^{a1j} & \cdots & T_d^{a1n} \\ \vdots & & \vdots & & \vdots \\ T_d^{ai1} & \cdots & T_d^{aij} & \cdots & T_d^{ain} \\ \vdots & & \vdots & & \vdots \\ T_d^{an1} & \cdots & T_d^{anj} & \cdots & T_d^{anm} \end{bmatrix} \tag{7}$$

where $T_d^{aij}$ is a submatrix of $T_d^a$. Next, $T_d^{a12}$ was taken as an example to calculate the submatrix as follows:

$$T_d^{a12} = \begin{bmatrix} t_{11}^{12}/t_1^{12} & \cdots & t_{1j}^{12}/t_1^{12} & \cdots & t_{1n_2}^{12}/t_1^{12} \\ \vdots & & \vdots & & \vdots \\ t_{i1}^{12}/t_i^{12} & \cdots & t_{ij}^{12}/t_i^{12} & \cdots & t_{in_2}^{12}/t_i^{12} \\ \vdots & & \vdots & & \vdots \\ t_{n_1 1}^{12}/t_{n_1}^{12} & \cdots & t_{n_1 j}^{12}/t_{n_1}^{12} & \cdots & t_{n_1 n_2}^{12}/t_{n_1}^{12} \end{bmatrix}, t_i^{12} = \sum_{j=1}^{n_2} t_{ij}^{12} \tag{8}$$

Step 7: The unweighted supermatrix $W$ was obtained.

The standardized total influence matrix obtained in the previous step was transposed.

$$W = (T_d^a)' \tag{9}$$

Step 8: The weighted supermatrix $W_a$ was calculated.

The standardized total influence matrix $T_D^a$ of the first-level dimension criteria was multiplied by the unweighted supermatrix $W$ of the second-level factor criteria.

$$W_a = T_D^a W = \begin{bmatrix} t_D^{a11} \times W_{11} & \cdots & t_D^{a1j} \times W_{1j} & \cdots & t_D^{a1n} \times W_{1n} \\ \vdots & & \vdots & & \vdots \\ t_D^{ai1} \times W_{i1} & \cdots & t_D^{aij} \times W_{ij} & \cdots & t_D^{ain} \times W_{in} \\ \vdots & & \vdots & & \vdots \\ t_D^{an1} \times W_{n1} & \cdots & t_D^{anj} \times W_{nj} & \cdots & t_D^{ann} \times W_{nn} \end{bmatrix} \tag{10}$$

$$T_D^a = \left[ t_D^{aij} \right]_{n \times n} = \begin{bmatrix} t_D^{11}/d_1 & \cdots & t_D^{1j}/d_1 & \cdots & t_D^{1n}/d_1 \\ \vdots & & \vdots & & \vdots \\ t_D^{i1}/d_i & \cdots & t_D^{ij}/d_i & \cdots & t_D^{in}/d_i \\ \vdots & & \vdots & & \vdots \\ t_D^{n1}/d_n & \cdots & t_D^{nj}/d_n & \cdots & t_D^{nn}/d_n \end{bmatrix}, d_i = \sum_{j=1}^{n} t_D^{ij}, i = 1, 2, 3, \ldots, n$$

Step 9: The weighted supermatrix $W_a$ was limited.

The weighted supermatrix was multiplied and the limit was calculated to obtain the stable-limit supermatrix $L$. The DANP weights were then found using

$$L = \lim_{\lambda \to \infty} (W_a)^\lambda = \begin{bmatrix} w_1 & w_1 & \cdots & w_1 \\ w_2 & w_2 & \cdots & w_2 \\ \vdots & \vdots & \vdots & \vdots \\ w_n & w_n & \cdots & w_n \end{bmatrix}_{n \times n} \tag{11}$$

## 4. Analysis Results

As mentioned in Section 2, the evaluation index system for assessing designers' willingness to implement green design practices involves the four dimensions of external motivation, corporate-level drivers, product-level drivers and designers' attributes, and a total of 18 indicators were categorized according to these four dimensions. To encourage designers to actively participate in green design projects, this study attempted to clarify the degree of the influence relationships between the dimensions and the indicators associated with each dimension and determine the key factors that influenced designers' willingness to participate in green design. The DANP model, which is explained in detail in Section 3, was used to explore the internal influence relationships between dimensions and the indicators associated with each dimension.

To implement a comprehensive evaluation, we surveyed 10 manufacturing companies in the city of Xiamen in China from 12 December 2021, to 26 December 2021. We interviewed the person in charge of the design department of each company. Basic information concerning the 10 companies is shown in Table 2. All 10 experts had more than8 years of experience in product design practice. A designer is usually someone who creates or produces creative work in a specific area of expertise. Design work usually covers aesthetics, technology, marketing and promotion. The sample of experts we selected was from 10 manufacturing enterprises in Xiamen, all of whom were the heads of the design departments of enterprises. These companies covered five industries: cultural creativity, household products, smart equipment, information services and mobile health technology. Cultural creativity and household products are driven by design aesthetics and marketing strategies, while mobile

health technology is based on demand mining. The starting point of intelligent equipment belongs to product development in the high-tech field and has high requirements for technical solutions. To sum up, the expert sample covered different industries that had a wide range of design attributes and a comprehensive understanding of the product life cycle. We interviewed each expert for approximately 25 min and spent approximately 30 min asking them to fill out the questionnaire. A questionnaire was designed for this study to evaluate the degree of influence between any two indicators according to Table 1. Experts were invited to respond to allow for pairwise comparisons of the degrees of influence between the indicators. As seen in Table A1 (Appendix A), an 18 × 18 average initial direct relation matrix was calculated by averaging 10 experts' responses. The consistency gaps in the 10 questionnaires were 3.5% according to Equation (2), which is smaller than 5%, and the confidence level was 97.22%, which is more than 95%. These results show good consistency and allow our results to reflect aspects of real situations.

**Table 2.** The background information of the 10 experts.

| Interviewees | Working Experience (Years) | Work Organization | Recently Involved Project |
|---|---|---|---|
| Design director | 16 | Vehicle design | Vehicle–road digital collaboration project |
| Design director | 17 | Robot education design | Kids' coding project |
| Design director | 12 | Bathroom design | Smart toilet project |
| Design director | 16 | Consumer electronics design | Wireless Bluetooth headset project |
| Design director | 15 | Consumer electronics design | Bicycle smart riding wear project |
| Design director | 15 | Health technology | Smart seat project |
| Design director | 14 | Smart manufacturing | Intelligent file management cabinet project |
| Design director | 8 | Cultural innovation | Stone tea tray design project |
| Design director | 11 | Home manufacturing | Bamboo tea table design project |
| Design director | 21 | Cultural innovation | Ceramic tea set items |

The total influence matrix for the dimensions and criteria within each dimension was calculated according to Equation (5), as seen in Tables 3 and A2. The values that reflect the sum of the influence given and received among criteria and dimensions was obtained as per step 5 of Section 3.

**Table 3.** Total influence matrix of dimensions.

| Dimensions | $D_1$ | $D_2$ | $D_3$ | $D_4$ |
|---|---|---|---|---|
| $D_1$ | 0.30 | 0.32 | 0.36 | 0.28 |
| $D_2$ | 0.25 | 0.26 | 0.31 | 0.24 |
| $D_3$ | 0.29 | 0.30 | 0.35 | 0.27 |
| $D_4$ | 0.27 | 0.29 | 0.33 | 0.26 |

As seen in Table 4, external motivation ($D_1$) was the most influential dimension because it had the largest net value (0.14). That is, external motivation had a powerful effect on the other three dimensions and was the key factor motivating designers to participate in green design practice. The largest ($r_i + c_i$) value (2.56) was for product-level drivers ($D_3$), which meant that this factor had the largest total influence degree among these four dimensions. To a certain extent, whether a designer can produce qualified green products is a reflection of the designer's professional level and work value. Therefore, the quality of the designer's design product was closely related to their participation in green design practice. The influential network relationship map was plotted according to Table 3, as shown in

Figure 2. As seen in Figure 2, external motivation ($D_1$) and designer attributes ($D_4$) both had a positive effect on corporate-level drivers ($D_2$) and product-level drivers ($D_3$), with external motivation ($D_1$) having a significant positive effect on the other three dimensions.

**Table 4.** Sum of the influence given and received according to the criteria and dimensions.

| Dimensions | $r_i$ | $c_i$ | $r_i + c_i$ | $r_i - c_i$ | Criteria | $r_i$ | $c_i$ | $r_i + c_i$ | $r_i - c_i$ |
|---|---|---|---|---|---|---|---|---|---|
| $D_1$ | 1.26 | 1.12 | 2.37 | 0.14 | $C_{11}$ | 6.12 | 4.12 | 10.24 | 2.00 |
| | | | | | $C_{12}$ | 5.93 | 5.14 | 11.07 | 0.79 |
| | | | | | $C_{13}$ | 5.60 | 5.19 | 10.79 | 0.40 |
| | | | | | $C_{14}$ | 5.62 | 5.70 | 11.32 | 0.08 |
| | | | | | $C_{15}$ | 5.01 | 5.21 | 10.21 | −0.20 |
| $D_2$ | 1.06 | 1.17 | 2.23 | −0.11 | $C_{21}$ | 4.60 | 5.30 | 9.90 | −0.70 |
| | | | | | $C_{22}$ | 4.82 | 5.22 | 10.04 | −0.40 |
| | | | | | $C_{23}$ | 4.88 | 5.46 | 10.34 | −0.58 |
| $D_3$ | 1.21 | 1.35 | 2.56 | −0.15 | $C_{31}$ | 5.67 | 6.12 | 11.79 | −0.45 |
| | | | | | $C_{32}$ | 5.62 | 6.05 | 11.67 | −0.43 |
| | | | | | $C_{33}$ | 5.81 | 6.30 | 12.12 | −0.49 |
| | | | | | $C_{34}$ | 4.47 | 6.08 | 10.54 | −1.61 |
| | | | | | $C_{35}$ | 5.58 | 6.22 | 11.80 | −0.64 |
| $D_4$ | 1.15 | 1.04 | 2.19 | 0.12 | $C_{41}$ | 5.48 | 4.52 | 10.01 | 0.96 |
| | | | | | $C_{42}$ | 5.43 | 5.11 | 10.55 | 0.32 |
| | | | | | $C_{43}$ | 5.14 | 5.30 | 10.44 | −0.17 |
| | | | | | $C_{44}$ | 4.39 | 4.10 | 8.48 | 0.29 |
| | | | | | $C_{45}$ | 5.52 | 4.53 | 10.05 | 0.98 |

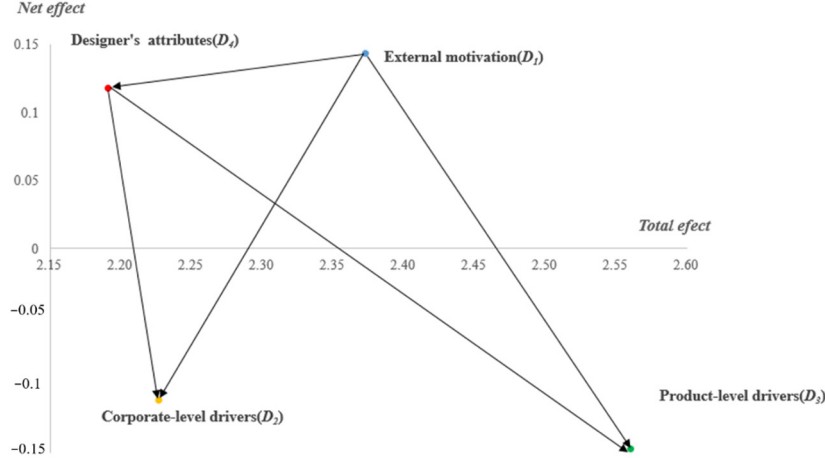

**Figure 2.** Influential network relationship map.

The weights of the dimensions and criteria were calculated according to Equation (6) to Equation (11), as seen in Table 5.

**Table 5.** Weights of the four dimensions and 18 criteria.

| Dimensions | Local Weight | Ranking | Criteria | Local Weight | Ranking | Global Weight | Ranking |
|---|---|---|---|---|---|---|---|
| $D_1$ | 0.239 | 3 | $C_{11}$ | 0.163 | 5 | 0.039 | 17 |
| | | | $C_{12}$ | 0.202 | 4 | 0.048 | 13 |
| | | | $C_{13}$ | 0.204 | 3 | 0.049 | 12 |
| | | | $C_{14}$ | 0.225 | 1 | 0.054 | 9 |
| | | | $C_{15}$ | 0.206 | 2 | 0.049 | 11 |
| $D_2$ | 0.250 | 2 | $C_{21}$ | 0.332 | 2 | 0.083 | 2 |
| | | | $C_{22}$ | 0.327 | 3 | 0.082 | 3 |
| | | | $C_{23}$ | 0.341 | 1 | 0.085 | 1 |
| $D_3$ | 0.289 | 1 | $C_{31}$ | 0.198 | 4 | 0.057 | 7 |
| | | | $C_{32}$ | 0.201 | 2 | 0.058 | 5 |
| | | | $C_{33}$ | 0.209 | 1 | 0.061 | 4 |
| | | | $C_{34}$ | 0.193 | 5 | 0.056 | 8 |
| | | | $C_{35}$ | 0.199 | 3 | 0.058 | 6 |
| $D_4$ | 0.222 | 4 | $C_{41}$ | 0.192 | 4 | 0.043 | 16 |
| | | | $C_{42}$ | 0.217 | 2 | 0.048 | 14 |
| | | | $C_{43}$ | 0.225 | 1 | 0.050 | 10 |
| | | | $C_{44}$ | 0.174 | 5 | 0.039 | 18 |
| | | | $C_{45}$ | 0.193 | 3 | 0.043 | 15 |

## 5. Discussion

According to Figure 2, external motivation ($D_1$) and designer attributes ($D_4$) were causal factors, and corporate-level drivers ($D_2$) and product-level drivers ($D_3$) were outcome factors. Thus, external motivation and designer attributes were the essential driving forces for implementing green design. Product-level drivers ($D_3$) had the largest total effect (2.56) and the smallest net effect (−0.15), which meant that product-level drivers were easily affected by other dimensions and indicators. Government policy, market demand, public recognition and other external factors affect development trends in green product design and popular preferences. The organization and management strategy of the enterprise, product design and product packaging are all affected by external factors. In addition, the designer's design level, educational background, design attitude and other personal factors are affected by external factors, but these aspects also affect the quality of product design and the company's development prospects. In addition to hiring outstanding designers for high-quality green products, companies must also pay attention to the country's policy orientation and market demand.

As illustrated in Table 4, regarding the external motivation dimension ($D_1$), government policy ($C_{11}$), market demand ($C_{12}$) and public recognition ($C_{13}$) influenced design trends ($C_{14}$) and industry organization support ($C_{15}$). The government advocates for the development and application of green and environmentally friendly technologies and guides the future development direction of the market. For example, the Chinese government has advocated vigorously for the development of green industries, such as wind power, solar energy and new energy vehicles, in recent years. The result is the promotion of the rapid development of the new energy industry. In terms of corporate-level drivers ($D_2$), the impact of the three indicators was weak. In terms of product-level drivers ($D_3$), the five indicators had weaker influence relationships, which might have been due to the fact that $D_3$ was the resulting factor and that these five indicators were more affected by $D_1$ and $D_4$. It was an interesting conclusion that a designer's abilities ($C_{41}$), personal norms and attitudes ($C_{42}$), and educational background ($C_{44}$) all had a positive influence on interpersonal relationships ($C_{43}$).

As seen in Table 5, product-level drivers ($D_3$, 0.289) were ranked first in terms of the weight, followed by corporate-level drivers ($D_2$, 0.250), external motivation ($D_1$, 0.239) and designer attributes ($D_4$, 0.222). According to the criteria, corporate reputation ($C_{23}$, 0.085), organizational strategy ($C_{21}$, 0.083) and institutional pressure ($C_{22}$, 0.082), which were

categorized as corporate-level drivers ($D_2$), were the three most important criteria. The five indicators of green materials and technology ($C_{33}$, 0.061), product standards ($C_{32}$, 0.058), overall design plan ($C_{35}$, 0.058), manufacturing process and surface treatment process ($C_{31}$, 0.057), and package design ($C_{34}$, 0.056), were all associated with the dimension of product-level drivers ($D_3$) and were ranked fourth through eighth. These results again demonstrated the findings of the DEMATEL analysis that corporate-level drivers ($D_2$) and product-level drivers ($D_3$) were significant outcome factors because the eight most significant criteria were related to $D_2$ and $D_3$.

The corporate strategy, reputation and corporate pressure of the designer's company are important factors that promote the designer's implementation of green design. Therefore, companies' incentive measures and care for employees are particularly important to the improvement of designers' work performance. Green materials and green technologies are important drivers for attracting designers' interest. The realization of green product standards and production processes are important indicators for designers' desire to participate in design considerations. It is worth noting that although the indicators involved in the external factors and the designer's own attributes were ranked low in terms of importance, this result did not mean that they were unimportant. On the one hand, the average weight of the 18 indicators was 0.056 and the standard deviation was 0.0002, which indicated that the distribution of weights among the 18 indicators was concentrated. On the other hand, the 10 experts were all executives from the design departments of manufacturing enterprises. It is understandable that they focused on the two dimensions of enterprise and product and employed a business perspective.

## 6. Conclusions

To identify the key driving factors that influence designers to implement green design practices, this study proposed an evaluation system that included four dimensions, and 18 criteria were selected based on the literature review. These dimensions included external motivation, corporate-level drivers, product-level drivers and designer attributes. A DANP method that combined the DEMATEL method and the ANP model was applied to determine the causation between and weights of these dimensions and criteria.

The results showed that product-level drivers and corporate-level drivers were the most important factors, accounting for 53.9% of the total weight, and that external motivation and designers' attributes accounted for 46.1% of the total weight. In fact, the weight distribution of the four dimensions was balanced, which meant that research concerning the drivers of designers' participation in green practices remains underdeveloped. Given the importance of a low-carbon economy, it is necessary to encourage designers to actively participate in green practices.

This study offers certain contributions. First, an evaluation system for identifying the driving factors that influence designers to implement green design practices was constructed. Second, a DANP model was applied to parse the evaluation system, and an impact diagram was drawn to show the causal relationship between any two dimensions. The key dimensions and criteria were explored to discover the motivations attracting designers to engage in green design practice.

This study also faced certain limitations. Ten experts from corporate design departments were interviewed, and their opinions represented the perspective of corporate design. There may be certain differences between their standpoints and academic perspectives. Fuzzy theory or rough numbers can be used to reduce the uncertainty regarding experts' judgments for further study.

**Author Contributions:** Conceptualization, investigation, resources, supervision, original draft preparation, project administration, funding acquisition, G.W.; methodology, C.J.; software, validation, formal analysis, data curation, writing—review and editing, Q.S.; visualization, supervision, J.J.H.L. All authors have read and agreed to the published version of the manuscript.

**Funding:** This research was funded by "Major Project of Fujian Social Science Research Base (FJ2018JDZ056), Ministry of Education Industry-university Cooperation Collaborative Education Project (202002100017) and Xiamen Education Science 'Thirteenth five-year Plan' 2017 Annual Project (1724)".

**Institutional Review Board Statement:** Not applicable.

**Informed Consent Statement:** Not applicable.

**Acknowledgments:** The authors are extremely grateful to the *Sustainability* journal's editorial team's valuable comments regarding improving the quality of this article.

**Conflicts of Interest:** The authors declare no conflict of interest.

## Appendix A

Detailed Results

**Table A1.** Initial direct influence matrix.

| | $C_{11}$ | $C_{12}$ | $C_{13}$ | $C_{14}$ | $C_{15}$ | $C_{21}$ | $C_{22}$ | $C_{23}$ | $C_{31}$ | $C_{32}$ | $C_{33}$ | $C_{34}$ | $C_{35}$ | $C_{41}$ | $C_{42}$ | $C_{43}$ | $C_{44}$ | $C_{45}$ |
|---|---|---|---|---|---|---|---|---|---|---|---|---|---|---|---|---|---|---|
| $C_{11}$ | 0.00 | 3.00 | 3.30 | 2.10 | 3.50 | 3.20 | 2.70 | 2.90 | 3.30 | 2.70 | 2.80 | 2.50 | 2.40 | 2.30 | 1.60 | 1.70 | 1.70 | 2.90 |
| $C_{12}$ | 2.80 | 0.00 | 2.80 | 3.30 | 2.70 | 2.60 | 2.40 | 2.00 | 2.80 | 2.90 | 3.20 | 3.10 | 2.90 | 1.50 | 2.20 | 2.30 | 1.50 | 2.00 |
| $C_{13}$ | 2.50 | 3.10 | 0.00 | 3.00 | 2.40 | 2.30 | 2.10 | 2.20 | 2.50 | 2.50 | 2.80 | 2.30 | 2.60 | 1.90 | 2.00 | 2.20 | 1.80 | 2.20 |
| $C_{14}$ | 1.70 | 2.90 | 2.20 | 0.00 | 2.30 | 2.30 | 1.90 | 1.60 | 2.60 | 3.10 | 3.10 | 3.00 | 3.30 | 2.20 | 2.60 | 2.40 | 1.80 | 1.70 |
| $C_{15}$ | 2.00 | 1.60 | 2.10 | 2.30 | 0.00 | 2.70 | 2.90 | 2.40 | 2.20 | 2.20 | 2.30 | 2.00 | 2.30 | 2.00 | 2.00 | 1.70 | 1.60 | 2.00 |
| $C_{21}$ | 1.30 | 1.40 | 1.60 | 1.70 | 2.70 | 0.00 | 2.90 | 2.30 | 2.30 | 2.20 | 2.10 | 2.10 | 2.10 | 1.60 | 1.70 | 1.90 | 1.30 | 2.10 |
| $C_{22}$ | 1.40 | 1.60 | 1.60 | 1.80 | 2.70 | 3.10 | 0.00 | 2.10 | 2.60 | 2.40 | 2.50 | 2.20 | 2.40 | 1.50 | 1.70 | 2.10 | 1.50 | 1.80 |
| $C_{23}$ | 1.40 | 1.70 | 1.80 | 1.80 | 2.20 | 2.60 | 2.60 | 0.00 | 2.80 | 2.30 | 2.60 | 2.50 | 2.20 | 1.40 | 1.60 | 2.00 | 2.30 | 1.70 |
| $C_{31}$ | 2.10 | 2.60 | 2.20 | 2.40 | 2.00 | 1.90 | 2.50 | 2.50 | 0.00 | 3.60 | 3.30 | 3.00 | 3.20 | 1.90 | 2.20 | 2.30 | 1.30 | 2.00 |
| $C_{32}$ | 1.50 | 2.40 | 2.20 | 2.90 | 2.10 | 2.00 | 2.10 | 2.70 | 3.30 | 0.00 | 3.60 | 3.20 | 3.20 | 1.80 | 2.30 | 2.50 | 1.10 | 1.80 |
| $C_{33}$ | 2.20 | 2.70 | 2.60 | 3.10 | 2.10 | 2.44 | 2.00 | 2.50 | 3.00 | 3.20 | 0.00 | 3.20 | 3.20 | 2.10 | 2.40 | 2.50 | 1.40 | 1.50 |
| $C_{34}$ | 1.30 | 2.30 | 2.30 | 2.40 | 1.40 | 1.50 | 1.70 | 1.70 | 2.30 | 2.20 | 2.30 | 0.00 | 2.10 | 1.70 | 1.90 | 1.90 | 1.40 | 1.50 |
| $C_{35}$ | 1.80 | 2.40 | 2.50 | 2.90 | 1.90 | 2.10 | 2.20 | 2.80 | 2.80 | 3.00 | 2.90 | 2.90 | 0.00 | 1.90 | 2.70 | 2.10 | 2.00 | 1.60 |
| $C_{41}$ | 1.20 | 1.80 | 2.00 | 2.20 | 2.10 | 2.20 | 2.00 | 2.30 | 2.30 | 2.50 | 2.70 | 2.60 | 2.70 | 0.00 | 3.20 | 3.10 | 2.50 | 2.60 |
| $C_{42}$ | 1.10 | 2.00 | 2.10 | 2.80 | 1.90 | 1.90 | 1.70 | 2.50 | 2.60 | 2.80 | 2.80 | 2.80 | 3.20 | 2.20 | 0.00 | 2.90 | 2.40 | 1.80 |
| $C_{43}$ | 1.20 | 1.90 | 2.00 | 2.00 | 1.90 | 1.80 | 1.90 | 2.50 | 2.60 | 2.20 | 2.40 | 2.40 | 2.90 | 2.50 | 2.60 | 0.00 | 2.50 | 2.00 |
| $C_{44}$ | 1.30 | 1.30 | 2.20 | 2.40 | 1.80 | 1.50 | 1.60 | 2.10 | 1.80 | 1.70 | 1.80 | 1.80 | 1.90 | 2.00 | 2.20 | 2.30 | 0.00 | 1.90 |
| $C_{45}$ | 2.70 | 2.20 | 2.20 | 2.10 | 2.40 | 2.50 | 2.70 | 2.60 | 2.70 | 2.20 | 2.60 | 2.30 | 2.50 | 2.20 | 2.00 | 2.60 | 1.50 | 0.00 |

**Table A2.** Total influence matrix of criteria in each dimension.

| | C11 | C12 | C13 | C14 | C15 | C21 | C22 | C23 | C31 | C32 | C33 | C34 | C35 | C41 | C42 | C43 | C44 | C45 |
|---|---|---|---|---|---|---|---|---|---|---|---|---|---|---|---|---|---|---|
| $C_{11}$ | 0.23 | 0.34 | 0.35 | 0.35 | 0.35 | 0.35 | 0.34 | 0.36 | 0.40 | 0.38 | 0.40 | 0.38 | 0.39 | 0.29 | 0.31 | 0.32 | 0.26 | 0.31 |
| $C_{12}$ | 0.28 | 0.27 | 0.33 | 0.37 | 0.33 | 0.33 | 0.32 | 0.33 | 0.38 | 0.38 | 0.40 | 0.38 | 0.39 | 0.27 | 0.31 | 0.33 | 0.25 | 0.28 |
| $C_{13}$ | 0.26 | 0.32 | 0.26 | 0.35 | 0.31 | 0.31 | 0.30 | 0.32 | 0.36 | 0.35 | 0.37 | 0.35 | 0.36 | 0.26 | 0.30 | 0.31 | 0.24 | 0.27 |
| $C_{14}$ | 0.24 | 0.32 | 0.31 | 0.29 | 0.31 | 0.31 | 0.30 | 0.31 | 0.36 | 0.37 | 0.38 | 0.37 | 0.38 | 0.27 | 0.31 | 0.31 | 0.24 | 0.26 |
| $C_{15}$ | 0.22 | 0.26 | 0.27 | 0.30 | 0.23 | 0.29 | 0.29 | 0.29 | 0.32 | 0.31 | 0.33 | 0.31 | 0.32 | 0.24 | 0.27 | 0.27 | 0.22 | 0.24 |
| $C_{21}$ | 0.20 | 0.24 | 0.25 | 0.27 | 0.27 | 0.22 | 0.27 | 0.27 | 0.30 | 0.29 | 0.30 | 0.29 | 0.30 | 0.22 | 0.24 | 0.26 | 0.19 | 0.23 |
| $C_{22}$ | 0.21 | 0.25 | 0.26 | 0.28 | 0.28 | 0.29 | 0.22 | 0.28 | 0.32 | 0.31 | 0.32 | 0.31 | 0.32 | 0.23 | 0.25 | 0.27 | 0.21 | 0.23 |
| $C_{23}$ | 0.21 | 0.26 | 0.26 | 0.28 | 0.27 | 0.28 | 0.28 | 0.24 | 0.32 | 0.31 | 0.33 | 0.31 | 0.31 | 0.23 | 0.25 | 0.27 | 0.22 | 0.23 |
| $C_{31}$ | 0.25 | 0.31 | 0.31 | 0.34 | 0.30 | 0.31 | 0.31 | 0.33 | 0.31 | 0.38 | 0.39 | 0.37 | 0.38 | 0.27 | 0.30 | 0.31 | 0.23 | 0.27 |
| $C_{32}$ | 0.24 | 0.31 | 0.31 | 0.35 | 0.30 | 0.31 | 0.30 | 0.33 | 0.37 | 0.30 | 0.39 | 0.37 | 0.38 | 0.26 | 0.30 | 0.32 | 0.23 | 0.26 |
| $C_{33}$ | 0.26 | 0.32 | 0.32 | 0.36 | 0.31 | 0.32 | 0.31 | 0.33 | 0.38 | 0.38 | 0.33 | 0.38 | 0.39 | 0.28 | 0.31 | 0.33 | 0.24 | 0.27 |
| $C_{34}$ | 0.19 | 0.25 | 0.25 | 0.28 | 0.24 | 0.24 | 0.24 | 0.25 | 0.29 | 0.29 | 0.30 | 0.24 | 0.29 | 0.22 | 0.24 | 0.25 | 0.19 | 0.21 |
| $C_{35}$ | 0.24 | 0.30 | 0.31 | 0.34 | 0.30 | 0.31 | 0.30 | 0.33 | 0.36 | 0.36 | 0.37 | 0.36 | 0.31 | 0.26 | 0.31 | 0.31 | 0.24 | 0.26 |
| $C_{41}$ | 0.23 | 0.29 | 0.29 | 0.32 | 0.30 | 0.30 | 0.30 | 0.31 | 0.35 | 0.35 | 0.36 | 0.35 | 0.36 | 0.22 | 0.32 | 0.32 | 0.25 | 0.27 |
| $C_{42}$ | 0.22 | 0.29 | 0.29 | 0.33 | 0.29 | 0.29 | 0.29 | 0.31 | 0.35 | 0.35 | 0.36 | 0.35 | 0.37 | 0.26 | 0.25 | 0.32 | 0.25 | 0.26 |
| $C_{43}$ | 0.21 | 0.27 | 0.28 | 0.30 | 0.28 | 0.28 | 0.28 | 0.30 | 0.33 | 0.32 | 0.34 | 0.33 | 0.34 | 0.26 | 0.29 | 0.24 | 0.24 | 0.25 |
| $C_{44}$ | 0.19 | 0.23 | 0.25 | 0.27 | 0.24 | 0.24 | 0.24 | 0.26 | 0.28 | 0.27 | 0.28 | 0.27 | 0.28 | 0.22 | 0.25 | 0.25 | 0.16 | 0.22 |
| $C_{45}$ | 0.26 | 0.30 | 0.30 | 0.32 | 0.31 | 0.31 | 0.31 | 0.32 | 0.36 | 0.34 | 0.36 | 0.35 | 0.36 | 0.27 | 0.29 | 0.31 | 0.23 | 0.22 |

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
