# Peer review of "Exploring the Driving Factors Influencing Designers to Implement Green Design Practices Based on the DANP Model"

_sustainability, doi:10.3390/su14116537_

Round 1

Reviewer 1 Report

Dear authors, thank you for your interesting papers, below some comments.

Your literature review focus on "green design" without taking into account other terms that describe the same/similar concept (ecodesign, sustainable design, environmental conscious design, etc.), some of these definition are currently widespread than green design. 

Did you decide to ignore them on the basis of some hypothesis? can you clearly state them or enrich the SoA looking at the additional definitions.

Despite DANP also allows to determine the relative importance of factors, the choice of the attributes to be analysed and how they are grouped have to be better justified (it is stated based on SoA, a table linking references and attributes will clarify these aspects.)

Section 2.2 Design definition is not underlying its technical and scientific parts, while the experts are coming also from technological sectors.

Table 1: check various errors. f.i. C23 definition equal to C32; in C25 standardized appears twice.

DANP decription is complete, a schema summarizing the various steps and their relationships can help the readers to better understanding the approach. References introducing DANP can be added.

Figure 1 better explain its content and teh implications about the dimensions' importance.

Expert opinion. do you use normal or geometric mean (geometric mean often used in approaches using AHP/ANP while dealing with multiple respondents)?

Various errors and some sections are not clear, advised a complete check.

Author Response

  1. Your literature review focus on "green design" without taking into account other terms that describe the same/similar concept (eco-design, sustainable design, environmental conscious design, etc.), some of these definition is currently widespread than green design.  Did you decide to ignore them on the basis of some hypothesis? can you clearly state them or enrich the so a looking at the additional definitions.

Answer:

Thank you for your helpful comments.

Eco-Design, green design, design for environment, environmental conscious design, and sustainable design meet the “green” characteristics of products in the whole product life cycle throughout the whole product of life. Sustainable design has a broader connotation, which requires not only the harmonious development of people and the environment, but also the sustainability of society, culture, consumption patterns, and lifestyles. We have added the concepts of green design in the Section 1.

  1. Despite DANP also allows to determine the relative importance of factors, the choice of the attributes to be analyzed and how they are grouped have to be better justified (it is stated based on SoA, a table linking references and attributes will clarify these aspects.)

Answer:

Thank you for your valuable suggestion. We have added the procedures how we decide the factors as follows:

Determining the relevant driving factors that motivate designers to engage in green design is the first step in developing and evaluating designers' green practice models. This paper uses literature reviews and experts’ discussions to determine the factors that affect designers’ green practices. Given the initial factors extracted from literature review, a survey was conducted to the experts to determine whether these factors are considered as essential factors for deriving the green design. Besides, they were asked to suggest additional deriving factors that are perceived as relevant according to their experience. After three rounds of discussions, four dimensions and 18 criteria were selected for the final evaluation. The specific dimensions and indicators are described in Table 1.

  1. Section 2.2 Design definition is not underlying its technical and scientific parts, while the experts are coming also from technological sectors.

Answer:

Thank you for your helpful comments. We have added the following explanations in section 2.2:

   From the perspective of technical application, green design involves renewable design technology, detachable technology design, and design technology for life cycle assessment. New products can be generated applying renewable design techniques to parts and materials of used or discarded products. The disassembly of the product is the premise of product recycling, which directly affects the recyclability of the product. How to conveniently and effectively evaluate the impact of products on people and the environment is the key to the adoption of green product design schemes by designers.

We also added the following explanations to strengthen the experts’ backgrounds in Section 4:

A designer is usually someone who creates or produces creative work in a specific area of expertise. Design work usually covers aesthetics, technology, marketing and promotion. The samples of experts were selected from 10 manufacturing enterprises in Xiamen who are the heads of the design departments of enterprises. These companies cover five industries: cultural creativity, household products, smart equipment, information services, and mobile health technology. Cultural creativity and household products are driven by design aesthetics and marketing strategies, while mobile health technology is based on demand mining. The starting point of the intelligent equipment belongs to the product development in the high-tech field, and has high requirements for technical solutions. To sum up, the expert sample covers different industries, who has a wide range of design attributes and comprehensive understanding of the product life cycle.

  1. Table 1: check various errors. f.x. C23 definition equal to C32; in C25 standardized appears twice.

Thank you for your helpful comments. We carefully revised the format errors in Table 1, and the revised content is as follows, marked in blue:

Table 1. Indicators and references for the green design evaluation framework.

Dimensions

Criteria

Explanation

External motivation (D1)

Government policy(C11)

The government formulates relevant green design policies to encourage enterprises to implement green design strategies.

Market  demand(C12)

Adopting sustainable green design can improve product competitiveness and market share.

Public recognition(C13)

Consumers’ conception of green consumption has been significantly improved, and they tend to buy products with green design attributes.

Design trends (C14)

Designers have increased their green design awareness and green design capabilities, and green design has become a design idea and design method to which the design industry has paid a great deal of attention.

Industry organizational support (C15)

The promotion and support of green design by associations, societies and other industry organizations and the recognition of green design via design awards.

Corporate-level drivers (D2)

Organizational strategy (C21)

The enterprise establishes a certain organizational structure and uses a systematic method to promote the implementation of green design within the enterprise.

Institutional pressure (C22)

The incorporation of the implementation of green design into project evaluation indicators.

Corporate reputation (C23)

Green design can enhance a company’s industry and social influence.

Product-level

drivers (D3)

Manufacturing process and surface treatment process (C31)

The manufacturing process and surface treatment process meets relevant environmental protection, requirements and qualify as a green manufacturing process.

Product standards (C32)

Products comply with market environmental standards and green design standards.

Green materials and technology (C33)

The use of environmentally friendly materials and technologies comply with relevant environmental requirements and green manufacturing processes.

Package design (C34)

The packaging uses environmentally friendly materials and technologies to reduce transportation and storage costs and to facilitate recycling.

Overall design plan (C35)

The overall design is scientific, standardized, and green.

Designer’s

attributes (D4)

Designers’ abilities (C41)

The accumulation of green design knowledge, methods and experience to enhance green design capabilities.

Personal norms and attitudes (C42)

Recognition of green design ideas and concepts and mastery of green design standards.

Interpersonal relationships (C43)

Designers extensively exchange green design concepts, methods, experiences, etc.

Legal awareness (C44)

Compliance with corresponding green specifications and environmental protection standards of national and international regulations.

Educational background (C45)

The designer’s level of education and knowledge of theories and methods related to green design.

  1. DANP description is complete, a schema summarizing the various steps and their relationships can help the readers to better understanding the approach. References introducing DANP can be added.

Thank you for your helpful comments. We have added a flow chart to illustrate the principle of DANP, as shown in Figure 1:

Figure 1. The process of data processing by the DANP method

We have added 4 papers about the interpretation of the DANP model:

[27] Rao, S.H. Transportation synthetic sustainability indices: A case of Taiwan intercity railway transport. Ecolo. Indic. 2021, 127, 107753.

[28] Jafari-Sadeghi, V., Mahdiraji, H.A., Busso, D., Yahiaoui, D. Towards agility in international high-tech SMEs: Exploring key drivers and main outcomes of dynamic capabilities. Technol. Forecast. Soc. 2022, 174, 121272.

[29] Apaydın, O., Aladağ, Z. Ranking the evaluation criteria of Hi-Fi audio systems and constricted information space: A novel method for determining the DEMATEL threshold value. Appl. Acoust. 2022, 190, 108584.

[30] Tsai, P.H., Wang, Y.W., Yeh, H.J. An evaluation model for the development of more humane correctional institutions: Evidence from Penghu Prison. Eval. Program. Plan. 2021, 89, 102013.

  1. Figure 1 better explain its content and the implications about the dimensions’ importance.

Thank you for your helpful comments. We have explained the implications about the dimensions’ importance in Section 5, as follows:

According to Figure 2, external motivation (D1) and designer attributes (D4) are causal factors, and corporate-level drivers (D2) and product-level drivers (D3) are outcome factors. Thus, external motivation and designer attributes are the essential driving forces to implement the green design. Product-level drivers (D3) have the largest total effect (2.56) and the smallest net effect (-0.15), which means that product-level drivers are easily affected by other dimensions and indicates. Government policy, market demand, public recognition and other external factors affect development trends in green product design and popular preferences. The organization and management strategy of the enterprise, product design and product packaging are all affected by external factors. In addition, the designer’s design level, educational background, design attitude and other personal factors are affected by external factors, but these aspects also affect the quality of product design and the company’s development prospects. In addition to hiring outstanding designers for high-quality green products, companies must also pay attention to the country’s policy orientation and market demand.

  1. Do you use normal or geometric mean (geometric mean often used in approaches using AHP/ANP while dealing with multiple respondents)?

Answer:

Thank you for your helpful comments. Yes, we used geometric mean to get the average matrix.

  1. Various errors and some sections are not clear, advised a complete check.

Answer:

Thank you for your helpful comments. We will revise the original text carefully, and the revised place will be marked in blue, please refer to the revised manuscript.

  1. Your literature review focus on "green design" without taking into account other terms that describe the same/similar concept (eco-design, sustainable design, environmental conscious design, etc.), some of these definition is currently widespread than green design.  Did you decide to ignore them on the basis of some hypothesis? can you clearly state them or enrich the so a looking at the additional definitions.

Answer:

Thank you for your helpful comments.

Eco-Design, green design, design for environment, environmental conscious design, and sustainable design meet the “green” characteristics of products in the whole product life cycle throughout the whole product of life. Sustainable design has a broader connotation, which requires not only the harmonious development of people and the environment, but also the sustainability of society, culture, consumption patterns, and lifestyles. We have added the concepts of green design in the Section 1.

  1. Despite DANP also allows to determine the relative importance of factors, the choice of the attributes to be analyzed and how they are grouped have to be better justified (it is stated based on SoA, a table linking references and attributes will clarify these aspects.)

Answer:

Thank you for your valuable suggestion. We have added the procedures how we decide the factors as follows:

Determining the relevant driving factors that motivate designers to engage in green design is the first step in developing and evaluating designers' green practice models. This paper uses literature reviews and experts’ discussions to determine the factors that affect designers’ green practices. Given the initial factors extracted from literature review, a survey was conducted to the experts to determine whether these factors are considered as essential factors for deriving the green design. Besides, they were asked to suggest additional deriving factors that are perceived as relevant according to their experience. After three rounds of discussions, four dimensions and 18 criteria were selected for the final evaluation. The specific dimensions and indicators are described in Table 1.

  1. Section 2.2 Design definition is not underlying its technical and scientific parts, while the experts are coming also from technological sectors.

Answer:

Thank you for your helpful comments. We have added the following explanations in section 2.2:

   From the perspective of technical application, green design involves renewable design technology, detachable technology design, and design technology for life cycle assessment. New products can be generated applying renewable design techniques to parts and materials of used or discarded products. The disassembly of the product is the premise of product recycling, which directly affects the recyclability of the product. How to conveniently and effectively evaluate the impact of products on people and the environment is the key to the adoption of green product design schemes by designers.

We also added the following explanations to strengthen the experts’ backgrounds in Section 4:

A designer is usually someone who creates or produces creative work in a specific area of expertise. Design work usually covers aesthetics, technology, marketing and promotion. The samples of experts were selected from 10 manufacturing enterprises in Xiamen who are the heads of the design departments of enterprises. These companies cover five industries: cultural creativity, household products, smart equipment, information services, and mobile health technology. Cultural creativity and household products are driven by design aesthetics and marketing strategies, while mobile health technology is based on demand mining. The starting point of the intelligent equipment belongs to the product development in the high-tech field, and has high requirements for technical solutions. To sum up, the expert sample covers different industries, who has a wide range of design attributes and comprehensive understanding of the product life cycle.

  1. Table 1: check various errors. f.x. C23 definition equal to C32; in C25 standardized appears twice.

Thank you for your helpful comments. We carefully revised the format errors in Table 1, and the revised content is as follows, marked in blue:

Table 1. Indicators and references for the green design evaluation framework.

Dimensions

Criteria

Explanation

External motivation (D1)

Government policy(C11)

The government formulates relevant green design policies to encourage enterprises to implement green design strategies.

Market  demand(C12)

Adopting sustainable green design can improve product competitiveness and market share.

Public recognition(C13)

Consumers’ conception of green consumption has been significantly improved, and they tend to buy products with green design attributes.

Design trends (C14)

Designers have increased their green design awareness and green design capabilities, and green design has become a design idea and design method to which the design industry has paid a great deal of attention.

Industry organizational support (C15)

The promotion and support of green design by associations, societies and other industry organizations and the recognition of green design via design awards.

Corporate-level drivers (D2)

Organizational strategy (C21)

The enterprise establishes a certain organizational structure and uses a systematic method to promote the implementation of green design within the enterprise.

Institutional pressure (C22)

The incorporation of the implementation of green design into project evaluation indicators.

Corporate reputation (C23)

Green design can enhance a company’s industry and social influence.

Product-level

drivers (D3)

Manufacturing process and surface treatment process (C31)

The manufacturing process and surface treatment process meets relevant environmental protection, requirements and qualify as a green manufacturing process.

Product standards (C32)

Products comply with market environmental standards and green design standards.

Green materials and technology (C33)

The use of environmentally friendly materials and technologies comply with relevant environmental requirements and green manufacturing processes.

Package design (C34)

The packaging uses environmentally friendly materials and technologies to reduce transportation and storage costs and to facilitate recycling.

Overall design plan (C35)

The overall design is scientific, standardized, and green.

Designer’s

attributes (D4)

Designers’ abilities (C41)

The accumulation of green design knowledge, methods and experience to enhance green design capabilities.

Personal norms and attitudes (C42)

Recognition of green design ideas and concepts and mastery of green design standards.

Interpersonal relationships (C43)

Designers extensively exchange green design concepts, methods, experiences, etc.

Legal awareness (C44)

Compliance with corresponding green specifications and environmental protection standards of national and international regulations.

Educational background (C45)

The designer’s level of education and knowledge of theories and methods related to green design.

  1. DANP description is complete, a schema summarizing the various steps and their relationships can help the readers to better understanding the approach. References introducing DANP can be added.

Thank you for your helpful comments. We have added a flow chart to illustrate the principle of DANP, as shown in Figure 1:

Figure 1. The process of data processing by the DANP method

We have added 4 papers about the interpretation of the DANP model:

[27] Rao, S.H. Transportation synthetic sustainability indices: A case of Taiwan intercity railway transport. Ecolo. Indic. 2021, 127, 107753.

[28] Jafari-Sadeghi, V., Mahdiraji, H.A., Busso, D., Yahiaoui, D. Towards agility in international high-tech SMEs: Exploring key drivers and main outcomes of dynamic capabilities. Technol. Forecast. Soc. 2022, 174, 121272.

[29] Apaydın, O., Aladağ, Z. Ranking the evaluation criteria of Hi-Fi audio systems and constricted information space: A novel method for determining the DEMATEL threshold value. Appl. Acoust. 2022, 190, 108584.

[30] Tsai, P.H., Wang, Y.W., Yeh, H.J. An evaluation model for the development of more humane correctional institutions: Evidence from Penghu Prison. Eval. Program. Plan. 2021, 89, 102013.

  1. Figure 1 better explain its content and the implications about the dimensions’ importance.

Thank you for your helpful comments. We have explained the implications about the dimensions’ importance in Section 5, as follows:

According to Figure 2, external motivation (D1) and designer attributes (D4) are causal factors, and corporate-level drivers (D2) and product-level drivers (D3) are outcome factors. Thus, external motivation and designer attributes are the essential driving forces to implement the green design. Product-level drivers (D3) have the largest total effect (2.56) and the smallest net effect (-0.15), which means that product-level drivers are easily affected by other dimensions and indicates. Government policy, market demand, public recognition and other external factors affect development trends in green product design and popular preferences. The organization and management strategy of the enterprise, product design and product packaging are all affected by external factors. In addition, the designer’s design level, educational background, design attitude and other personal factors are affected by external factors, but these aspects also affect the quality of product design and the company’s development prospects. In addition to hiring outstanding designers for high-quality green products, companies must also pay attention to the country’s policy orientation and market demand.

  1. Do you use normal or geometric mean (geometric mean often used in approaches using AHP/ANP while dealing with multiple respondents)?

Answer:

Thank you for your helpful comments. Yes, we used geometric mean to get the average matrix.

  1. Various errors and some sections are not clear, advised a complete check.

Answer:

Thank you for your helpful comments. We will revise the original text carefully, and the revised place will be marked in blue, please refer to the revised manuscript.

Reviewer 2 Report

It is not clear whether the combining of DEMATEL and ANP to create DANP is the authors' construction, or whether there are references to previous attempts to combine these methods? If it is in fact the authors who have created the method, it should be emphasised both in the title, abstract and text (under 3. Methodology for example).

The sample of 10 experts seems relatively small to draw clear conclusions from, as there are a number of confounding factors and bias that may be involved. However the evaluation system constructed appears to hold much potential and as a pilot project to test this system, the study has value.

Author Response

  1. It is not clear whether the combining of DEMATEL and ANP to create DANP is the authors’ construction, or whether there are references to previous attempts to combine these methods? If it is in fact the authors who have created the method, it should be emphasized both in the title, abstract and text (under 3. Methodology for example).

Thank you for your helpful comments. The hybrid model is derived and referred from some references. We have added 4 papers about the interpretation of the DANP model:

[27] Rao, S.H. Transportation synthetic sustainability indices: A case of Taiwan intercity railway transport. Ecological Indicators. 2021, 127, 107753.

[28] Jafari-Sadeghi, V., Mahdiraji, H.A., Busso, D., Yahiaoui, D. Towards agility in international high-tech SMEs: Exploring key drivers and main outcomes of dynamic capabilities. Technological Forecasting and Social Change. 2022, 174, 121272.

[29] Apaydın, O., Aladağ, Z. Ranking the evaluation criteria of Hi-Fi audio systems and constricted information space: A novel method for determining the DEMATEL threshold value. Applied Acoustics. 2022, 190, 108584.

[30] Tsai, P.H., Wang, Y.W., Yeh, H.J. An evaluation model for the development of more humane correctional institutions: Evidence from Penghu Prison. 2021, 89, 102013.

  1. The sample of 10 experts seems relatively small to draw clear conclusions from, as there are a number of confounding factors and bias that may be involved. However, the evaluation system constructed appears to hold much potential and as a pilot project to test this system, the study has value.

Thank you for your helpful comments. Due to limited time and insufficient financial support, we interviewed managers of the design departments of 10 companies. The sample may not be representative. There are explanations in the research limitations:

This study also faces certain limitations. Ten experts from corporate design departments were interviewed, and their opinions represented the perspective of corporate design. There may be certain differences between their standpoints and the academic perspectives. Fuzzy theory or rough number can be used to reduce uncertainty with respect to experts’ judgments for further study.

Reviewer 3 Report

none

Author Response

Thank you for your helpful comments.

Round 2

Reviewer 1 Report

Dear authors,

thank for the new version that I think better underline the results of your work.

I will probably still provide additional explanation in section 5 to clarify the direction of arrows in Figure 2 and how practitioners can use this. But I think is already understandable.

Some minor problems:

line 164 DRIVING instead of deriving

Table 1 STANDARDIZED is duplicated for factor C35

about line 238 some equations table seem superimposed

Author Response

Dear Co-Editor and Referees,

Thank you very much for your comments, we have revised the three errors according to your comments. We will appreciate it very much if this paper can be published in this highly esteemed journal.

Yours sincerely

James Liou, Professor

National Taipei University of Technology

No. 1 Chung-Hsaio E. Road Section 3

Taipei, 106, Taiwan

e-mail: jamesjhliou@gmail.com; jhliou@ntut.edu.tw